# Duration and Influencing Factors of Postoperative Urinary Incontinence after Robot-Assisted Radical Prostatectomy in a Japanese Community Hospital: A Single-Center Retrospective Cohort Study

**DOI:** 10.3390/ijerph20054085

**Published:** 2023-02-24

**Authors:** Tadashi Kasai, Taro Banno, Kazutaka Nakamura, Yukiko Kouchi, Haruki Shigeta, Fumio Suzuki, Yudai Kaneda, Divya Bhandari, Anju Murayama, Katumori Takamatsu, Naomi Kobayashi, Toyoaki Sawano, Yoshitaka Nishikawa, Hiroyuki Sato, Akihiko Ozaki, Tomohiro Kurokawa, Norio Kanzaki, Hiroaki Shimmura

**Affiliations:** 1Department of Rehabilitation, Jyoban Hospital of Tokiwa Foundation, Iwaki 972-8322, Japan; 2Department of Urology, Jyoban Hospital of Tokiwa Foundation, Iwaki 972-8322, Japan; 3School of Medicine, Tohoku University, Sendai 980-8575, Japan; 4School of Medicine, Hokkaido University, Sapporo 060-8638, Japan; 5Department of Breast and Thyroid Surgery, Jyoban Hospital of Tokiwa Foundation, Iwaki 972-8322, Japan; 6Medical Governance Research Institute, Tokyo 108-0074, Japan; 7Department of Surgery, Jyoban Hospital of Tokiwa Foundation, Iwaki 972-8322, Japan; 8Department of Health Informatics, Kyoto University School of Public Health, Kyoto 606-8501, Japan

**Keywords:** prostatectomy, robotics, prostatic neoplasms, urinary incontinence

## Abstract

Objectives: Post-operative urinary incontinence (PUI) after robotic-assisted radical prostatectomy (RARP) is an important complication; PUI occurs immediately after postoperative urethral catheter removal, and, although approximately 90% of patients improve within one year after surgery, it can significantly worsen their quality of life. However, information is lacking on its nature in community hospital settings, particularly in Asian countries. The purposes of this study were to investigate the time required to recover from PUI after RARP and to identify its associated factors in a Japanese community hospital. Methods: Data were extracted from the medical records of 214 men with prostate cancer who underwent RARP from 2019 to 2021. We then calculated the number of days elapsed from the surgery to the initial outpatient visit confirming PUI recovery among the patients. We estimated the PUI recovery rate using the Kaplan–Meier product limit method and evaluated associated factors using the multivariable Cox proportional hazards model. Results: The PUI recovery rates were 5.7%, 23.4%, 64.6%, and 93.3% at 30, 90, 180, and 365 days following RARP, respectively. After an adjustment, those with preoperative urinary incontinence experienced significantly slower PUI recovery than their counterparts, while those with bilateral nerve sparing experienced recovery significantly sooner than those with no nerve sparing. Conclusion: Most PUI improved within one year, but a proportion of those experiencing recovery before 90 days was smaller than previously reported.

## 1. Introduction

Prostate cancer is the third most common cancer globally, with 1,414,259 diagnosed cases in 2020 [1]. Among various treatment methods for prostate cancer, the main treatment measure has been surgery, namely radical prostatectomy. Robot-assisted radical prostatectomy (RARP) in particular has become the standard procedure, accounting for the majority of cases undergoing radical prostatectomy [2].

One of the most important postoperative complications of radical prostatectomy is PUI, which transiently but surely jeopardizes the postoperative quality of life (QOL) of patients [3,4]. While PUI is reportedly milder with RARP compared with open and laparoscopic radical prostatectomy, it is still an important complication that would exacerbate the QOL of prostate cancer patients; thereby, its management holds significant clinical implications [5].

PUI after radical prostatectomy occurs immediately after removal of the urethral catheter post-surgery. Recovery occurs over time, with approximately 90% of patients improving within 1 year after surgery [6]. It has been reported that various preoperative factors are associated with PUI, such as old age, high obesity, the presence of complications, preoperative erectile dysfunction, a short membranous urethral length, urethral volume, urethral morphology, and bladder factors including the presence of preoperative voiding muscle overactivity and poor bladder compliance [7,8,9,10]. For surgical methods, urethral sphincter-sparing and nerve-sparing as well as newly invented hood techniques are effective in preventing PUI [11]. In addition, it has been indicated that pelvic floor muscle exercises performed preoperatively stimulate early recovery from PUI [12].

In Japan, the number of prostate cancer diagnoses has been on the rise as the aging population increases. In 2018, prostate cancer had the highest number of patients among males, with 92,021 annual cases [13]. Moreover, the number of deaths has been increasing, with 12,759 in 2020, recording the highest number [14]. An improved prognosis for prostate cancer has further emphasized the clinical significance of proper PUI management following RARP, the predominant surgical procedure for prostate cancer. However, previous evidence of PUIs has been mostly based on research performed in Europe and the United States, with only limited reports available from Asia. In addition, Japanese clinical research on the duration of PUI has progressed mostly in university hospitals, and information is lacking on PUI management in general community hospitals. This is an important perspective given that RARP has been widely preformed outside of university hospitals, at least in Japan. Therefore, we aimed to investigate the duration of PUI after RARP at Jyoban Hospital, a community hospital that has conducted a large number of RARP procedures, and its associated factors.

## 2. Methods

### 2.1. Setting and Participants

This investigation was conducted at Jyoban Hospital of Tokiwa Foundation in Iwaki City, Hamadori Region of Fukushima Prefecture. The population of Iwaki City was approximately 320,000 as of October 2019. It has traditionally been regarded as a remote area, suffering from a physician undersupply in the long term: specifically, its number of medical doctors was 167 per 100,000 population in 2018, compared to the Japanese national average of 247 per 100,000 in the same year; and the average age of medical doctors was 56.4 years old in 2018, compared to the national average of 49.9 years old in the same year. In these difficult circumstances, the Tokiwa Foundation took over the operation of Jyoban Hospital in 2010 from Iwaki City, and the hospital has developed over time during the last decade. During this process, its urology department has taken the lead, expanding into one of the largest community-based urological departments in Japan at present. Indeed, the Department of Urology now has the latest version of the Da Vinci operation system, Da Vinci Xi (Intuitive Surgical Inc., Sunnyvale, CA, USA), and conducts various types of robot-assisted laparoscopic surgery, including RARP. In 2018, the department saw the hospitalization of 638 patients with prostate cancer, the fourth highest number in the country for prostate cancer treatment, with 113 RARPs performed in 2019.

In this study, we considered the patients who underwent RARP from 1 April 2019 to 31 March 2021. When considering the detailed procedure, an indication for nerve sparing was made separately for the left and right sides. Nerve sparing was performed for low- and intermediate-risk patients, according to D’Amico’s classification, on the side where no cancer was detected on biopsy or MRI. Along with this principle, we took the patient’s wishes into consideration when making a comprehensive decision on whether nerve sparing could be performed. Further, lymph node dissection (LND) was not performed in most of our patients. This was because there is little evidence that lymph node dissection in prostate cancer could provide additional benefits to the patient receiving surgery, and it would rather increase the risk of lower-extremity edema. In this sense, while lymph node dissection would lead to accurate staging, the direct therapeutic benefit is unknown as it is associated with poor perioperative outcomes [15]. In our institution, the procedure was performed by multiple surgeons, i.e., a primary surgeon and two or three assistants (according to chart data), and it may be performed by an experienced surgeon or by residents under the guidance of experienced surgeons.

### 2.2. Data Extraction

From the medical records of Jyoban Hospital, we extracted data on the following: the dates of RARP and initial outpatient visits confirming RARP recovery; age; presence or absence of type 2 diabetes; history of alcohol consumption and smoking; presence or absence of transurethral prostatic surgery for benign prostatic hyperplasia; presence or absence of preoperative radiotherapy; presence or absence of preoperative urinary incontinence; systolic and diastolic blood pressure; height; weight at surgery; body mass index (BMI); obesity, defined as body mass index of 25 or above; albumin level; initial prostate-specific antigen (PSA); preoperative Gleason score; D’Amico’s classification; pathological T stage; presence or absence of lymph node dissection; main operator; presence or absence of nerve sparing; postoperative complication of inguinal hernia and intestinal obstruction; presence or absence of continued pelvic floor muscle exercises.

The duration of PUI was defined as the number of days elapsed from the date when the RARP was conducted to the date of the shortest outpatient visit when the physician in charge confirmed the recovery from PUI among those considered. We defined PUI recovery as when the two following conditions were met: (1) the patient was aware that their PUI had improved and (2) they changed their urinary incontinence pads less than or equal to 1 pad per day [3]. The patients who used incontinence pads but did not change them were considered to have improved their PUI because they may have used them as precautionary measures. If the records of the degree of PUI diverged between a doctor and a nurse, the one with the lower grade was selected. Patients for whom the date of the outpatient visit could not be verified and patients for whom both the number of urinary incontinence pad changes and the number of PUI could not be verified were excluded.

### 2.3. Analysis Method

We conducted two analyses in this study. First, we estimated the rate of PUI recovery following the RARP using the Kaplan–Meier product limit method. Then, we constructed a Cox proportional hazard regression model for PUI recovery to evaluate its associated factors. We considered all the sociodemographic and clinical variables as covariates, using the backward stepwise variable selection method (inclusion criteria, *p*  <  0.1). The covariates with a small number of participants were re-grouped, as necessary. As a sensitivity analysis, we employed a multiple imputation method to fill in missing values for all the covariates. Based on an assumption of missing at random, we constructed the model 10 times using a Markov chain Monte Carlo method and integrated the results. All the data were analyzed with Stata version 15.0 (College Station, TX, USA).

## 3. Results

A total of 214 patients underwent RARP, and the analysis was performed on 209 patients after excluding five patients with missing values in the outcome (i.e., the time interval between surgery and urinary continence).

Sociodemographic and clinical patient information is shown in Table 1. The median age of the patients was 71 years (interquartile range 67–76), 11.5% (N = 24) had diabetes, 9.6% (N = 20) had preoperative urinary incontinence, and their median BMI was 24.4 (interquartile range 22.2–26.2), with 85 patients (40.7%) being obese. The median value of albumin and initial PSA were 3.9 g/dL (interquartile range 3.7–4.1) and 8.9 ng/mL (interquartile range 5.9–16.0). The most common preoperative Gleason score was 7, with a proportion of 45.0% (N = 94); 48.8% of the patients were diagnosed as high risk according to D’Amico’s classification before RARP, and 84.7% of the patients were diagnosed as pathological T2 after RARP. Further, 64.1% (N = 134) of the patients were operated on by experienced doctors, and unilateral and bilateral nerve sparing was achieved in 45.7% (N = 91) and 5.5% (N = 11) of the patients, respectively. Postoperative complications of inguinal hernia and intestinal obstructions occurred in 7.7% (N = 16) and 2.4% (N = 5) of the patients. Lastly, pelvic muscle exercise was performed in 93.8% (N = 195) of the patients.

Figure 1 shows the results of the Kaplan–Meier survival analysis curve. The rates of urinary continence were evaluated at 30 days (4 weeks/1 month), 90 days (12 weeks/3 months), 180 days (24 weeks/6 months), and 365 days (48 weeks/12 months), with recovery rates of 5.7%, 23.4%, 64.6%, and 93.3%, respectively.

Table 2 shows the results of univariable and multivariable Cox proportional hazards regression analysis. After an adjustment, those with preoperative urinary incontinence experienced significantly slower PUI recovery than those without preoperative urinary incontinence (hazard ratio 0.28, 95% confidence interval 0.14–0.57). Patients with high albumin levels had slower recovery from PUI than those with low albumin levels (hazard ratio 0.54, 95% confidence interval 0.35–0.81). In addition, recovery from PUI was slower after surgery performed by residents compared to surgery performed by experienced physicians (hazard ratio 0.61, 95% confidence interval 0.44–0.86). In contrast, those with bilateral nerve sparing experienced PUI recovery significantly sooner than those with no nerve sparing (hazard ratio 2.87, 95% confidence interval 1.43–5.77), while those with unilateral nerve sparing also tended to experience PUI recovery sooner than those with no nerve sparing (hazard ratio 1.35, 95% confidence interval 0.97–1.87). The sensitivity analysis using the multiple imputation method did not converge and we could not obtain reasonable findings.

## 4. Discussion

In this study investigating PUI recovery following RARP in a community hospital in a remote area suffering from a physician undersupply in Japan, we primarily found that rates of PUI recovery at 90 days (12 weeks/3 months) and 365 days (48 weeks/12 months) were 23.4% and 93.3%. We also found that preoperative urinary incontinence, higher albumin levels and surgery performed by unexperienced surgeons were associated with delayed PUI recovery, while nerve sparing was significantly associated with early recovery. This study predominantly presents important knowledge to medical institutions in a similar rural community setting in Japan, but, in the meantime, we believe that its implications could be valuable beyond this setting.

With regard to PUI recovery at 365 days, the obtained findings were no worse than the previous findings reported in the systematic review by Ficarra et al., where the proportion of PUI recovery at 12 months ranged from 89 to 92% [3]. We may have overestimated the proportion of PUI recovery in this study, given that our definition of PUI recovery was liberal, allowing the use of one pad per day, while some previous studies implemented a definition of no pads per day as the definition [3]. Nonetheless, it is notable that our observation was superior to that of all previous studies implementing a definition of no more than one pad per day for PUI recovery in the same systematic review [3]. In this respect, it is reasonable to say that at least we may have achieved outcomes comparable to the previous studies at 365 days.

In contrast, our findings of PUI recovery at 90 days (23.4%) were inferior to the figures reported in previous studies. Indeed, Ficarra et al. reported in their systematic review that the proportion of patients experiencing three-month PUI recovery was 65% [3]. It is difficult to conclusively determine the underlying mechanism of the disparity in the findings of ours and the previous studies. However, our observations about nerve sparing could provide important clues to understand this phenomenon. In our study, nerve sparing was associated with early PUI recovery. However, the proportion of patients with bilateral nerve sparing was relatively low at only 5.5%, primarily due to the high-risk profile of the patients, with only 11.0% classified as low risk according to D’Amico’s classification. A contrasting phenomenon was observed in a previous study by Kim et al., which showed that 53.9% (285/529) of the patients experienced bilateral nerve sparing and that 60% of them experienced PUI recovery by 12 weeks [16]. Further, given that the effect of nerve sparing appeared to have been strong during the earlier phase of PUI recovery [16], the low proportion of bilateral nerve sparing may have been a primary contributor to the delayed PUI recovery.

In addition, the presence of preoperative symptoms of urinary incontinence may have delayed the PUI recovery in this study. Ficarra et al. reported that preoperative lower urinary tract symptoms delayed the recovery from PUI [3]. However, given that only a limited proportion of the patients experienced preoperative urinary incontinence in this study, its contribution to delayed PUI recovery may have been limited.

In our study, surgery performed by inexperienced surgeons led to delayed PUI recovery, which is in line with multiple previous studies. While the acceptance of young doctors and their on-the-job training is a critical factor allowing hospitals located in rural settings to sustain their workforce and provide necessary care for local residents, it is also important to minimize the drawbacks resulting from this process. Jyoban Hospital recently implemented the dual console of the Da Vinci surgical system, which allows a senior doctor to provide real-time supervision of an operation conducted by younger surgeons.

Moreover, the patients with higher albumin levels had a longer recovery time from PUI, which has not been reported in the previous literature [17]. Furthermore, this finding differs from what one would intuitively expect from surgical findings regarding wound healing and requires further investigation.

It is also important to note any potential implications of PUI recovery following RARP in rural community settings. Indeed, our finding at 12 months was rather superior to that observed in a Japanese elite university hospital: Hakozaki et al. reported that only 85.0% of their patients experienced PUI recovery one year after RARP [18]. This means that rather than the status and ranking of hospitals, the experience and proficiency of surgeons affects the outcomes of patients, which is a fact explained above.

However, rural community hospitals face clear disadvantages compared to university hospitals, such as the challenge of organizing a comprehensive support framework for patients that involves multiple hospital staff. In this study, almost all patients performed preoperative pelvic floor muscle exercises, but no early improvement was seen. One reason for this is that we could not assign a single instructor to each patient but rather had three (or more) instructors working on rotation; therefore, the instructions for the exercises were not consistent. In addition, the instructions were mainly oral explanations; thus, the patients’ pelvic floor muscular contraction during the exercise was not accurately confirmed. Furthermore, in elderly patients, their understanding of the exercise could have been insufficient. These factors may have contributed to the inadequate effectiveness of the exercise, and we believe that a lack of manpower prevented us from providing better training to PUI. However, it has been demonstrated that pelvic floor muscle exercises are clinically significant [19,20,21], particularly in the early phase after surgery [12]. Thus, it is desirable to improve the way in which we teach pelvic floor muscle exercises so that every patient can enjoy their benefits. For example, the use of pelvic floor muscle exercise pamphlets could be useful in explaining the purposes of standardizing and unifying the content of instruction by physical therapists.

## 5. Limitations

There were several limitations in this study. First, this was a single-institution study with only a limited number of patients. This could have limited the generalizability of the observed findings and resulted in the omission of some important factors in the regression analysis, such as obesity [22], but this is the first study investigating PUI recovery after RARP in a Japanese community setting, which is an important novelty of the study. Second, we did not evaluate various confounding factors, such as anatomical ones (prostate size, preoperative urethral length, and maximum urethral closure pressure) and surgical procedure methods. As a result, the findings of the regression analysis may have been limited. Third, the definition of PUI recovery relying on the self-reported count of the urine pads may be affected by various biases. For example, the usage of urine pads may have not been standardized among the patients, which may have affected the counts of the pads used among the patients.

## 6. Conclusions

In this study, which examined the recovery from PUI following RARP in a Japanese community setting, we found that PUI was eliminated in 93.3% of the patients at 365 days, which was comparable to previous reports. However, recovery at 90 days was observed in only 23.4% of the patients, which was slower than reported in previous studies. Our analysis revealed that preoperative urinary incontinence, higher albumin levels and surgery performed by inexperienced surgeons were associated with a delayed recovery from PUI. On the other hand, nerve sparing was significantly associated with an earlier recovery from PUI.

## Figures and Tables

**Figure 1 ijerph-20-04085-f001:**
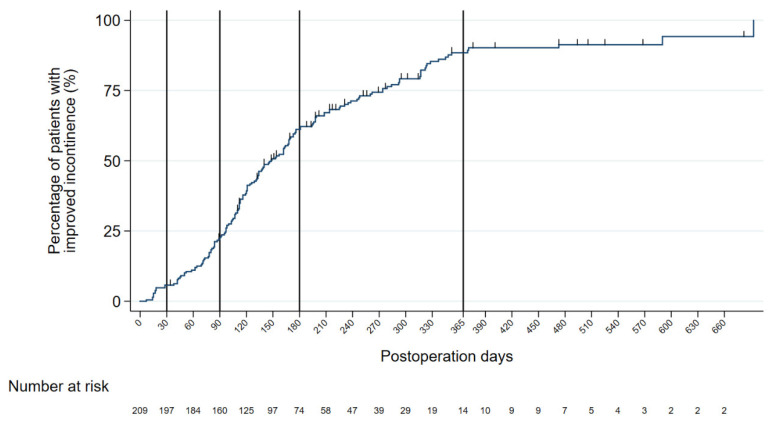
Recovery rate of postoperative urinary incontinence estimated with the Kaplan–Meier product limit method.

**Table 1 ijerph-20-04085-t001:** Patients’ characteristics.

Variable	All (N = 209)
**Age (median, interquartile range)**	71 (67–76)
**Diabetes (N, %)**	24 (11.5)
**Regular drinking (N, %) ^†a^**	121 (58.2)
**Regular smoking (N, %) ^†b^**	151 (73.0)
**History of transurethral prostatic surgery for benign prostatic hyperplasia (N, %)**	
HoLEP	10 (4.8)
TUR-P	2 (1.0)
**Preoperative radiation therapy (N, %)**	2 (1.0)
**Preoperative urinary incontinence (N, %)**	20 (9.6)
**Hypertension (defined as systolic blood pressure of 130 mmHg or above or diastolic blood pressure of 85 mmHg or above) (N, %)**	26 (12.4)
**Body mass index (median, interquartile range)**	24.4 (22.2–26.2)
**Obesity (defined as body mass index of 25 or above) (N, %)**	85 (40.7)
**Albumin (g/dL, median, interquartile range) ^‡c^**	3.9 (3.7–4.1)
**Initial prostate-specific antigen (ng/mL) (median, interquartile range)**	8.9 (5.9–16.0)
**Preoperative Gleason score (N, %)**	
6	26 (12.4)
7	94 (45.0)
8	38 (18.2)
9	45 (21.5)
10	6 (2.9)
**D’Amico’s classification (N, %)**	
Low risk	23 (11.0)
Intermediate risk	84 (40.2)
High risk	102 (48.8)
**Surgery by experienced doctors (N, %)**	134 (64.1)
**Nerve sparing (N, %) ^†d^**	
Bilateral nerve sparing	11 (5.5)
Unilateral nerve sparing	91 (45.7)
No nerve sparing	97 (48.7)
**Lymph node dissection (N, %)**	3 (1.4)
**Pathological T stage (N, %)**	
T0	13 (6.2)
T2	177 (84.7)
T3	18 (8.6)
T4	1 (0.5)
**Postoperative complication of inguinal hernia (N, %)**	16 (7.7)
**Postoperative complication of intestinal obstruction (N, %)**	5 (2.4)
**Pelvic floor muscle exercise (N, %) ^†e^**	195 (93.8)

^†^ Proportion was calculated excluding missing values; ^a^ this variable was missing in one patient; ^b^ this variable was missing in two patients; ^‡^ calculated excluding missing values; ^c^ this variable was missing in two patients; ^d^ this variable was missing in ten patients; ^e^ this variable was missing in one patient; HoLEP=holmium laser enucleation of the prostate; TUR-P=transurethral resection of the prostate.

**Table 2 ijerph-20-04085-t002:** Cox proportional hazard regression model for recovery from postoperative urinary incontinence without multiple imputation.

Variables	Unadjusted Hazard Ratio (95% CI)	Adjusted Hazard Ratio (95% CI)
**Age (years)**		
70 years old or less	Reference	
Over 70 years	0.78 (0.57–1.06)	
**Diabetes**		
No	Reference	
Yes	0.78 (0.48–1.27)	
**Regular drinking**		
No	Reference	
Yes	1.19 (0.88–1.63)	
**Regular smoking**		
No	Reference	
Yes	0.95 (0.67–1.34)	
**History of transurethral prostatic surgery for benign prostatic hyperplasia**		
No	Reference	
HoLEP	1.03 (0.51–2.10)	
TUR-P	0.50 (0.07–3.60)	
**Preoperative radiation therapy**		
No	Reference	
Yes	1.25 (0.31–5.05)	
**Preoperative urinary incontinence**		
No	Reference	
Yes	0.34 (0.18–0.65) ***	0.28 (0.14–0.57) ***
**Hypertension (defined as systolic blood pressure of 130 mmHg or above or diastolic blood pressure of 85 mmHg or above)**		
No	Reference	
Yes	1.16 (0.74–1.80)	
**Obesity (defined as body mass index of 25 or above)**		
No	Reference	
Yes	0.80 (0.58–1.08)	
**Albumin (g/dL** **)**	0.68 (0.46–1.00) *	0.54 (0.35–0.81) ***
**Initial prostate-specific antigen (ng/mL)**	1.00 (1.00–1.00)	
**Preoperative Gleason score**		
6	Reference	
7	0.90 (0.56–1.46)	
8 or above	0.67 (0.41–1.09)	
**D’Amico classification**		
Low/medium risk	Reference	
High risk	0.72 (0.53–0.98) **	
**Lymph node dissection**		
No	Reference	
Yes	0.76 (0.19–3.09)	
**Surgery by experienced doctors**		
Senior	Reference	
Resident	0.66 (0.48–0.91) **	0.61 (0.44–0.86) ***
**Nerve sparing**		
No nerve sparing	Reference	Reference
Unilateral nerve sparing	1.52 (1.10–2.09) **	1.35 (0.97–1.87) *
Bilateral nerve sparing	2.07 (1.06–4.04) **	2.87 (1.43–5.77) ***
**Pathological T stage**		
T2 or below	Reference	
T3 or above	0.89 (0.53–1.49)	
**Postoperative complication of hernia**		
No	Reference	
Yes	0.64 (0.36–1.15)	
**Postoperative complication of intestinal obstruction**		
No	Reference	
Yes	0.59 (0.22–1.60)	
**Pelvic floor muscle exercise**		
No	Reference	
Yes	1.38 (0.71–2.71)	

* *p* < 0.10, ** *p* < 0.05, *** *p* < 0.01; CI = confidence interval.

## Data Availability

The datasets analysed during the current study are available from the corresponding author on reasonable request.

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
