# Peer review of "Duration and Influencing Factors of Postoperative Urinary Incontinence after Robot-Assisted Radical Prostatectomy in a Japanese Community Hospital: A Single-Center Retrospective Cohort Study"

_ijerph, 2023, doi:10.3390/ijerph20054085_

Round 1
Reviewer 1 Report
This is an acceptably written article, which analyses the resolution of postoperative urinary incontinence after prostatectomy in prostate cancer.
I miss a better definition of postoperative urinary incontinence, as well as the methods of evaluation during follow-up, as this may lead to a significant bias that invalidates the results of the study.
Furthermore, from a structural point of view, the tables are difficult to read and understand. Both should be simplified, avoiding duplication of information (e.g. avoid duplication of YES/NO data in qualitative variables in table 1, limiting to only listing the YES data).
Author Response
Reviewer 1
I miss a better definition of postoperative urinary incontinence, as well as the methods of evaluation during follow-up, as this may lead to a significant bias that invalidates the results of the study.
Reply:
Thank you for your advice. Regarding the evaluation of urinary incontinence, in this case, we are evaluating the number of urinary incontinence pads changes. In a city hospital with a small number of medical staff, it is difficult to spend time on evaluation, so we think it is appropriate to use a simple number of pad exchanges for evaluation. We have clarified this point in the updated manuscript as follows.
(Page 3, lines 23 to 33 in the revised manuscript)
“The duration of PUI was defined as the number of days elapsed from the date when the RARP was conducted to the date of the shortest outpatient visit when the physician in charge confirmed the recovery of PUI among those considered. We defined the PUI recovery when the two following conditions were met: 1) the patient was aware that their PUI had improved and; 2) if they changed their urinary incontinence pads less than or equal to 1 pad per day.[3] The patients who used incontinence pads but did not change them were considered to have improved their PUI because they may have used them as precautionary measures. If the records of the degree of PUI diverged between a doctor and a nurse, the one with a lower grade was selected. Patients for whom the date of the outpatient visit could not be verified and patients for whom both the number of urinary incontinence pad changes and the number of PUI could not be verified were excluded.”
Furthermore, from a structural point of view, the tables are difficult to read and understand. Both should be simplified, avoiding duplication of information (e.g. avoid duplication of YES/NO data in qualitative variables in table 1, limiting to only listing the YES data).
Reply:
Thank you for your suggestion. In all tables, we have simplified them as you have indicated for ease of reading and understanding.
Reviewer 2 Report
Thank you very much for the opportunity to review the manuscript.
The authors performed a retrospective cohort study regarding postoperative urinary incontinence after RARP in a Japanese community hospital.
I have several critiques and/or suggestions for the authors.
Major points
1) Instead of Lines 76-83, In the nature of perioperative functional outcomes, the authors must provide information on the detailed routine procedure (including an indication of NS and/or LND) as well as surgeon (single surgeon? well-experience? or resident with supervision?)
2) I do not understand why the rates of bilateral nerve-sparing were so high despite the rates of high-risk patients being more than 50%. Please clarify. In addition, why the early incontinence recovery rate was so low despite the such a high rate of bilateral nerve-sparing? Please discuss it compared to published results.
3)In the statistical methods of multiple imputations, the authors only described two lines (Lines 117-118). How many datasets did the author create using multiple imputations, and how integrate the results of Cox regression analysis? Please clarify.
4) In the analysis of the Cox proportional hazard regression model (Table 2), why did the authors not include the nerve-sparing as a covariate of multivariable analysis? I believe this variate is one of the essential factors affecting the recovery of PUI (Bilateral vs. Non-nerve sparing, as well). I would recommend to reanalyze using this as variates are included in multivariable analysis.
5) Discussion is immature and does not reflect the results of this study. For example, the first three paragraphs regarding BMI, D’Amico, and nerve-sparing are only general information not from this study (all variates were negative for PUI recovery in multivariable analysis). Specifically for BMI, in this Japanese cohort, BMI was obviously lower than in other countries, such as the cohort that the author cited (15). In addition, the authors failed to show obesity as a prognosticator of continence. In addition, the aim of this study was “to investigate the duration of PUI after RARP at Joban Hospital, a community hospital with a large number of RARP, as it is an important perspective given that the RARP has been widely performed outside of the university hospitals at least in Japan.”? So, the authors need to discuss the differences between your results and previously published results from university/academic/ high-volume hospitals.
I suggest some points to improve the discussion
・I suggest that the author need to use “we found” or “our study revealed” in the sentence of the results from the present study.
・Line 180 needs a reference
・Line 190-192: These novel findings need to be discussed more, such as wound healing, as the author mentioned.
・Line 208: This must be clearly described in the methods or results section.
・Line 209-210: I do not believe this is a primary reason. And the statement is strong. Please rewrite and/or use "One possible reason," "we hypothesize," or "seems to".
Minor but essential points
1) The authors should briefly define the PUI in the Abstract
2) In Figure 1, one of the main results of this study, values are somewhat different between the sentence and KM curve. the values in Figure are all lower than that in the text. The results are unreliable.
3)In Lines 102-106, I did not find 2)
4)Terminology: Following the EAU, NCCN guidelines, “total prostatectomy” should be “radical prostatectomy”; therefore, I recommend using “RARP (robot-assisted radical prostatectomy)
Author Response
Reviewer #2
Methological:
Instead of Lines 76-83, In the nature of perioperative functional outcomes, the authors must provide information on the detailed routine procedure (including an indication of NS and/or LND) as well as surgeon (single surgeon? well-experience? or resident with supervision?)
Reply:
We thank the Peer Reviewers for their suggestions. We have updated the manuscript to fully reflect the Peer Reviewer's responses. We have added the following detailed perioperative information on nerve sparing and lymphatic dissection as well as details about the surgeon. We have also added references regarding lymph node dissection.
(Page 2, line 48 to page 3, line 10 in the revised manuscript)
“Indications for nerve sparing are considered separately for the left and right sides, and nerve sparing is performed in low- and intermediate-risk patients according to D'Amico's classification on the side where no cancer is detected on biopsy or no findings of cancer on MRI is pointed out. Finally, we listen to the patient's wishes and make a comprehensive decision on whether or not nerve sparing should be performed. Lymph node dissection (LND) is not performed in most of our patients. This is because there is little evidence that lymph node dissection in prostate cancer could provide additional benefits to the patients receiving surgeries, and rather it would increase a risk of lower extremity edema. In this sense, while lymph node dissection would lead to accurate staging, the direct therapeutic benefit is unknown as it is associated with poor perioperative outcomes [15]. In our institution, the procedure was performed by multiple surgeons: a primary surgeon and two or three assistants (according to chart data), and it may have been performed by an experienced surgeon or by residents under guidance of experiences surgeons.”
Reference  
15)  Fossati, N., et al., The Benefits and Harms of Different Extents of Lymph Node Dissection During Radical Prostatectomy for Prostate Cancer: A Systematic Review. Eur Urol, 2017. 72(1): p. 84-109.
I do not understand why the rates of bilateral nerve-sparing were so high despite the rates of high-risk patients being more than 50%. Please clarify.
Reply:
Thank you for pointing this out. We have reviewed the data again and found an error in the nerve preservation data. In the data used in the original submission, unilateral preservation was incorrectly categorized as complete preservation. In this revision, we have now corrected this error, correctly and only categorizing bilateral preservation as complete preservation. As a result, a rate of bilateral nerve preservation has been updated into 5.5% (11 patients). Therefore, it can be said that a rate of patients experiencing bilateral nerve preservation is low. We have now updated the main text and Table accordingly.
(Page 4, lines 8 to 10 in the revised manuscript)
“Further, 64.1% (N=134) of the patients were operated by experienced doctors and unilat-eral and bilateral nerve sparing was achieved in 45.7% (N=91) and 5.5% (N=11) of the pa-tients, respectively.”
(Page 7, lines 27 to 28 in the revised manuscript)
“However, the proportion of the patients with bilateral nerve sparing was relatively low only at 5.5%,”
In addition, why the early incontinence recovery rate was so low despite the such a high rate of bilateral nerve-sparing? Please discuss it compared to published results.
Reply:
As mentioned previously, we have reviewed the data and have now found that a proportion of the patients with bilateral nerve preservation was low (5.5%, 11 cases). Previous studies have shown that a high percentage of bilateral nerve sparing may correlate with faster urinary incontinence recovery time. For example, Kim et al. reported a high rate of bilateral nerve preservation of over 50% (53.9%, 285/529) and urinary incontinence recovery of over 60% (66.0%, 12w/3mo) in 12 weeks [16].Compared to previous studies, the lower rate of bilateral nerve preservation may have contributed to the delayed duration of urinary incontinence recovery. We have now revised the text based on the above information and added new comparative literature.
(Page 7, lines 26 to 35 in the revised manuscript)
“In our study, nerve sparing was associated with an early PUI recovery. However, the proportion of the patients with bilateral nerve sparing was relatively low only at 5.5%, primarily due to the high-risk profile of the patients, with only 11.0% classified as low risk according to Damico’s classification. A contrasting phenomenon was observed in a previous study by Kim et al., which showed that 53.9% (285/529) of the patients experienced bilateral nerve sparing and that 60% of them experienced a PUI recovery by 12 weeks [16].Further, given that an effect of nerve sparing appeared to have been strong during an earlier phase of PUI recovery [16], a low proportion of bilateral nerve sparing may have been a primary contributor of the delayed PUI recovery.”
Reference  
16)  Kim, M., et al., Integrity of the Urethral Sphincter Complex, Nerve-sparing, and Long-term Continence Status after Robotic-assisted Radical Prostatectomy. Eur Urol Focus, 2019. 5(5): p. 823-830.
In the statistical methods of multiple imputations, the authors only described two lines (Lines 117-118). How many datasets did the author create using multiple imputations, and how integrate the results of Cox regression analysis? Please clarify.
Reply:
Thank you for your question. First, we excluded 5 patients with missing values in the outcome (e.g., the time interval between surgery and urinary continence). Then, we employed multiple imputation methods for the remaining 209 patients: based on an assumption of the Missing at random, we calculated 10 times in a Markov chain Monte Carlo method and integrated the results. To address this point, we have revised the manuscript accordingly.
(Page 3, lines 35 to 44 in the revised manuscript)
“We conducted two analyses in this study. First, we estimated the rate of PUI recovery at a key time point following the RARP using Kaplan-Meier product-limit methods. Then, we constructed a Cox proportional hazard regression models for the PUI recovery to evaluate its associated factors. We considered all the sociodemographic and clinical variables as covariates, using the backward stepwise variable selection method (inclusion criteria, p < 0.1). The covariates with a small number of participants were re-grouped, as necessary. As a sensitivity analysis, we employed a multiple imputation method to fill in missing values for all the covariates. Based on an assumption of Missing at random, we constructed the model 10 times using a Markov chain Monte Carlo method and integrated the results.”
In the analysis of the Cox proportional hazard regression model (Table 2), why did the authors not include the nerve-sparing as a covariate of multivariable analysis? I believe this variate is one of the essential factors affecting the recovery of PUI (Bilateral vs. Non-nerve sparing, as well). I would recommend to reanalyze using this as variates are included in multivariable analysis.
Reply:
Thank you for pointing this out. In the original submission, we did consider nerve-sparing as a covariate of the regression model. However, the variable did not remain in the final model as a result of the variable selection process. In any case, since data on nerve-sparing included an error, we have updated the text and tables, respectively. In the revised paper, nerve sparing remained in the final model as a variable significantly associated with a timing of PUI recovery. As a result, those with bilateral nerve sparing experienced a PUI recovery significantly sooner than those with no nerve sparing (hazard ratio 2.87, 95% confidence interval 1.43–5.77), while those with unilateral nerve sparing also tended to experience a PUI recovery sooner than those with no nerve sparing (hazard ratio 1.35, 95% confidence interval 0.97–1.87).
(Page 4, lines 9 to 10 in the revised manuscript)
“unilateral and bilateral nerve sparing was achieved in 45.7% (N=91) and 5.5% (N=11) of the patients, respectively.”
(Page 5, lines 19 to 23 in the revised manuscript)
“those with bilateral nerve sparing experienced a PUI recovery significantly sooner than those with no nerve sparing (hazard ratio 2.87, 95% confidence interval 1.43–5.77), while those with unilateral nerve sparing also tended to experience a PUI recovery sooner than those with no nerve sparing (hazard ratio 1.35, 95% confidence interval 0.97–1.87).”
Discussion is immature and does not reflect the results of this study. For example, the first three paragraphs regarding BMI, D’Amico, and nerve-sparing are only general information not from this study (all variates were negative for PUI recovery in multivariable analysis). Specifically for BMI, in this Japanese cohort, BMI was obviously lower than in other countries, such as the cohort that the author cited (15). In addition, the authors failed to show obesity as a prognosticator of continence.
Reply:
Thank you. While obesity tended to delay a continence following a surgery in a univariate analysis (hazard ratio 0.80, 95% confidence interval 0.58–1.08), it did not remain in the final model probably because of limited number of patients. To address this point, we have updated the limitations section as follows.
(Page 8 Lines 41 to 44 in the updated manuscript)
“This could have limited a generalizability of the observed findings and result in the omission of some important factors in the regression analysis such as obesity [22], but this is the first study investigating PUI recovery after RARP in a Japanese community setting, which is an important novelty of the study.”
Reference  
22)  Sarychev, S., et al., Impact of obesity on perioperative, functional and oncological outcomes after robotic-assisted radical prostatectomy in a high-volume center. World J Urol, 2022. 40(6): p. 1419-1425.
In addition, the aim of this study was “to investigate the duration of PUI after RARP at Joban Hospital, a community hospital with a large number of RARP, as it is an important perspective given that the RARP has been widely performed outside of the university hospitals at least in Japan.”? So, the authors need to discuss the differences between your results and previously published results from university/academic/ high-volume hospitals.
Reply:
Thank you for pointing this out. When we compared the results of this survey with those from university/academic/large hospitals, the results were generally better than those from university hospitals in the long term, although the recovery period was slower in the early years.This seems to indicate that the experience and proficiency of the surgeon affects patient outcomes more than the status or rank of the hospital. In Japan, the da Vinci has become remarkably popular, and many RARP surgeries are performed at general hospitals in the city. Therefore, although there is a limitation of a single-center survey, we believe that the results of this study are extremely useful.
(Page 8, lines 17 to 22 in the revised manuscript)
“It is also important to argue any potential implications of PUI recovery following RARP in rural community settings. Indeed, our finding at 12 month was rather superior to that observed in a Japanese elite university hospital: Hakozaki et al. reported that only 85.0% of the patients experienced a PUI recovery one year after RARP [18]. This means that rather than a status and rank of hospitals, experiences and proficiency of surgeons would affect outcome of the patients, which is a fact explained above.”
Reference  
18)  Hakozaki, K., et al., Predictors of urinary function recovery after laparoscopic and robot-assisted radical prostatectomy. Int Braz J Urol, 2023. 49(1): p. 50-60.
I suggest some points to improve the discussion
I suggest that the author need to use “we found” or “our study revealed” in the sentence of the results from the present study.
Reply:
Thank you for your advice. We will change the subject to "we found" in the manuscript.
Line 180 needs a reference
Reply:
Many thanks for your advice. In this revision, we have carefully cited the necessary references throughout the manuscript.
Line 190-192: These novel findings need to be discussed more, such as wound healing, as the author mentioned.
Reply:
Thank you for your advice. While an inverse association between post-operative urinary incontinence recovery and albumin level is a novel finding, it is impossible to conclusively decipher its mechanism just with available evidence. Still, we discussed this point in the discussion as follows.
(Page 8, lines 13 to 16 in the revised manuscript)
“Moreover, patients with higher albumin levels had a longer recovery time from PUI, which has not been reported in previous literature [17]. Furthermore, this finding differs from what one would intuitively expect from surgical findings regarding wound healing and requires further investigation.”
Reference
17)  Shao, I.H., et al., Predictors of short-term and long-term incontinence after robot-assisted radical prostatectomy. J Int Med Res, 2018. 46(1): p. 421-429.
Minor but essential points
The authors should briefly define the PUI in the Abstract
Reply:
We have included a brief definition of PUI in the abstract in this updated.
(Page 1, lines 19 to 22 in the revised manuscript)
“Post-operative urinary incontinence (PUI) after robotic-assisted radical prostatectomy (RARP) is an important complication; PUI occurs immediately after postoperative urethral catheter removal and, although approximately 90% of patients improve within one year after surgery, it can significantly worsen quality of life.”
In Figure 1, one of the main results of this study, values are somewhat different between the sentence and KM curve. the values in Figure are all lower than that in the text. The results are unreliable.
Reply:
Thank you for your suggestion. We have reviewed the extracted data and performed the analysis again. In the revised manuscript, we have corrected the values with the text and KM curve and changed Figure 1 to the new one.
In Lines 102-106, I did not find 2)
Reply:
Thank you for pointing this out. Some numbers were missing, and we have corrected and restated them.
(Page 3, lines 25 to 28 in the revised manuscript)
“We defined the PUI recovery when the two following conditions were met: 1) the patient was aware that their PUI had improved and; 2) if they changed their urinary incontinence pads less than or equal to 1 pad per day.[3]”
Reference  
3)  Ficarra, V., et al., Systematic review and meta-analysis of studies reporting urinary continence recovery after robot-assisted radical prostatectomy. Eur Urol, 2012. 62(3): p. 405-17.
Terminology: Following the EAU, NCCN guidelines, “total prostatectomy” should be “radical prostatectomy”; therefore, I recommend using “RARP (robot-assisted radical prostatectomy)
Reply:
The reference to RALP in the text has been corrected to RARP (robot-assisted radical prostatectomy).
Round 2
Reviewer 2 Report
The authors well revised point-by-point according to reviewer's suggestions.
The quality of this paper is much improved.